# Stumpy forms are the predominant transmissible forms of *Trypanosoma brucei*

**Jean Marc Tsagmo Ngoune[1], Parul Sharma[1,2], Aline Crouzols[1], Nathalie Petiot[1], Brice Rotureau[1,3]\***

[1]Trypanosome Transmission Group, Trypanosome Cell Biology Unit, Institut Pasteur, Université Paris Cité, Paris, France; [2]Sorbonne Université, ED515 Complexité du Vivant, Paris, France; [3]Parasitology Unit, Institut Pasteur of Guinea, Conakry, Guinea

## eLife Assessment

African (or Salivarian) trypanosomes are significant pathogens of humans and domestic animals. For many decades is was accepted that only the "stumpy" non-proliferative form was capable of infecting the Tsetse-fly vector, but recent work challenged this, suggesting that the proliferative "slender" form is also infective. The current paper provides **important** and **convincing** laboratory evidence that the original concept is probably correct for most infections: the slender form was not infective for adult Tsetse, and was only able to infect young, less immunocompetent flies if N-acetyl glucosamine was added to the feed.

**\*For correspondence:**
rotureau@pasteur.fr

**Competing interest:** The authors declare that no competing interests exist.

**Abstract** Schuster et al. demonstrated that bloodstream slender forms of African trypanosomes are readily transmissible to young tsetse flies where they can complete their complex life cycle (Schuster et al., 2021). In their experimental conditions, a single slender parasite was sufficient for productive infection. Here, we compared the infectivity of slender and stumpy bloodstream forms in adult flies with a mature immune system, and without using any chemical compounds that would alter the insect immune response and/or promote the infection. After ingestion of slender forms, infected flies were observed only in 1 out of 24 batches of non-immunocompetent teneral flies and with a high number of parasites. In contrast, infected flies were detected in 75% (18/24) of the batches infected with stumpy parasites, and as few as 10 stumpy parasites produced mature infections in immune adult flies. We discuss that, although Schuster et al. have demonstrated the intrinsic capacity of slender form trypanosomes to infect young and naive tsetse flies, highlighting the remarkable plasticity and adaptability of these protists, this phenomenon is unlikely to significantly contribute to the epidemiology of African trypanosomiases. According to both experimental and field observations, stumpy forms appear to be the most adapted forms for African trypanosome transmission from the mammalian host to the tsetse fly vector in natural conditions.

## Introduction

Protist parasites of the *Trypanosoma brucei* group cause human African trypanosomiasis (HAT), or sleeping sickness in humans, and nagana in cattle (*Büscher et al., 2017*). They are transmitted by the blood-feeding tsetse fly following a long (at least 3 weeks) and complex (at least nine distinct stages) cyclical development (review in *Rotureau and Van Den Abbeele, 2013*). In the mammalian host's blood circulation, proliferating slender trypanosomes differentiate into cell cycle-arrested stumpy cells upon quorum sensing when they reach high parasite densities (*Vassella et al., 1997*; *Dean et al.,*

**eLife digest** Human African Trypanosomiasis – also known as sleeping sickness – is a deadly disease caused by the single-celled parasite *Trypanosoma brucei*. The parasite has a complex life cycle that involves both humans and tsetse flies. When a tsetse fly bites an infected human, it can take up the parasite and pass it on to other people.

Inside the human host, *T. brucei* exists in two forms: a rapidly dividing 'slender' form, and a non-proliferative 'stumpy' form that emerges once a high enough density of parasitic cells has been reached. For decades, scientists thought that only the stumpy form can successfully spread from humans to tsetse flies. However, a 2021 study challenged this view, suggesting that the slender form might also be transmissible.

To investigate this further, Ngoune et al. examined whether, and under what conditions, the slender and stumpy forms of *T. brucei* could infect tsetse flies. The team grew both forms of the parasite in the laboratory and fed them to tsetse flies in an environment designed to resemble natural conditions. The midgut and salivary glands of the flies were then dissected four weeks later to assess the level of infection.

Ngoune et al. found that slender forms of the parasite were only able to infect one out of 24 batches of young tsetse flies – and only when each fly ingested up to 10,000 parasites. The slender forms also failed to infect adult flies entirely, likely because they have a more robust immune system. In contrast, stumpy forms of the parasite where much more readily transmitted, successfully infecting about 75% of all the tested fly batches, even when as few as 10 parasitic cells were ingested.

The study by Ngoune et al. reaffirms the longstanding view that the stumpy form of *T. brucei* is the primary stage at which the parasite is transmitted from humans to flies. While the slender form of *T. brucei* may be capable of infecting tsetse flies under certain conditions, these results suggest that it rarely makes this jump and is therefore unlikely to play a significant role in the spread of sleeping sickness.

*2009*; *Mony et al., 2014*; *Rojas et al., 2019*). This differentiation is thought not only to regulate the parasite load in the reservoir host (*Turner et al., 1995*), but also to provide transmissible parasites adapted to pursue the life cycle in the vector host (*Rico et al., 2013*). Indeed, stumpy forms express several transcripts and proteins necessary to the next developmental stage in the insect, the procyclic form, including the Protein Associated to Differentiation 1 or PAD1 (*Dean et al., 2009*). For decades, the arrest of the cell cycle and differentiation to the stumpy stage were presumed essential for the developmental progression of bloodstream trypanosomes to the insect stages.

Recently, Schuster et al. demonstrated that slender trypanosomes can also present some intrinsic characteristics of transmissible forms (PAD1 mRNAs and proteins) and are readily transmissible to both young male and female tsetse flies, where they can complete their complex life cycle (*Schuster et al., 2021*), yet with a lower efficiency than stumpy forms (*Matthews and Larcombe, 2022*). In their experimental conditions, a single slender parasite was sufficient for productive infection. These laboratory conditions are, however, significantly different from what is encountered in the field. First, only young teneral flies (1–3 days post-eclosion) with an immature immune system were used. Second, in some experiments, chemical compounds altering the insect immune response (glutathione) and/or promoting the infection (*N*-acetyl-glucosamine) were added to the infective meal. To assess the importance of these parameters, we challenged the infectivity of slender bloodstream forms in adult tsetse flies, i.e., in conditions closer to the natural situation.

## Results

Pleomorphic *T. b. brucei* bloodstream forms were either maintained in culture at a density lower than $5 \cdot 10^5$ parasites/ml to prevent quorum-sensing-induced differentiation and obtain only slender forms or induced for differentiation with a 5'-AMP nucleotide analogue to obtain mostly stumpy forms. The expression of PAD1 at the cell surface was assessed by immunofluorescence analysis prior to each experimental infection: no PAD1 expression was detected in the slender group, whereas an average of 63% (52–71%, n=12 replicates) of the induced cells were expressing PAD1. Batches of 50 teneral

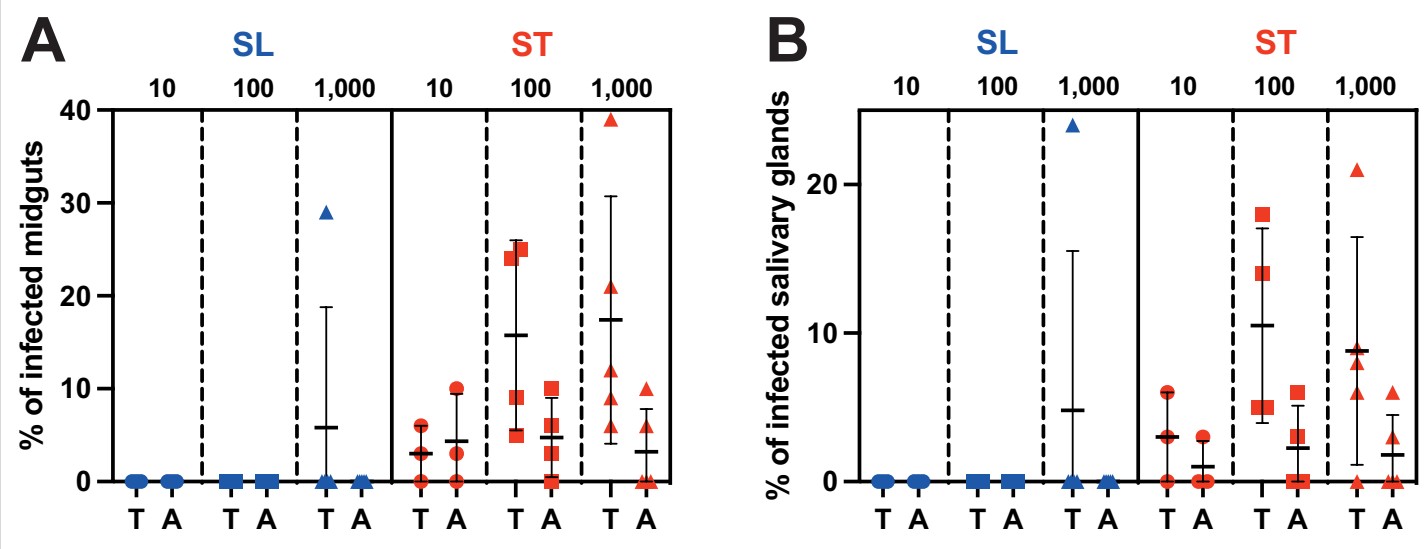

**Figure 1.** Stumpy forms are the predominant transmissible forms. Comparison of (**A**) midgut and (**B**) salivary gland infection rates in teneral (**T**) vs. adult (**A**) tsetse flies (batches of 50 flies) infected with 10–100 (circles, three independent experiments), 100–1000 (squares, four independent experiments), or 1000–10,000 (triangles, five independent experiments) parasites in the slender (SL in blue) or stumpy (ST in red) forms.

The online version of this article includes the following source data for figure 1:

**Source data 1.** Source data used to create *Figure 1*.

(<72 hr) or adult (2–3 weeks) male tsetse flies were fed in parallel with either slender or stumpy forms at densities corresponding to individual ingestion of about 10, 100, or 1000 parasites per blood meal. In total, 1384 flies from 12 distinct experimental infections were dissected about 4 weeks (28–31 days) after parasite ingestion. Infection rates in midguts and salivary glands were quantified and plotted for each condition (*Figure 1*, *Figure 1—source data 1*).

After ingestion of slender forms, infected flies were observed in only 1 batch out of 24. This occurred in not yet fully immunocompetent teneral flies and with the highest number of ingested parasites (1000–10,000 parasites). In contrast, midgut and salivary glands infected flies were observed in 75% (18/24) and 62.5% (15/24) of the batches infected with stumpy parasites, respectively. As few as 10 stumpy parasites produced mature infections in immunocompetent adult flies, and the infection rates were similar whatever the amounts of stumpy forms ingested. In more susceptible nonimmune teneral flies, the infection rates were increasing with the number of stumpy forms ingested.

Differences between the strain clones, the cell culture conditions, and/or the fly colony maintenance conditions could explain part of the differences in infection rates observed here as compared to the *Schuster et al., 2021*, study. Nevertheless, the use of the lectin-inhibitory sugar *N*-acetyl-glucosamine to enhance infection rates in the latter study could be a more likely explanation. To assess this hypothesis, an additional

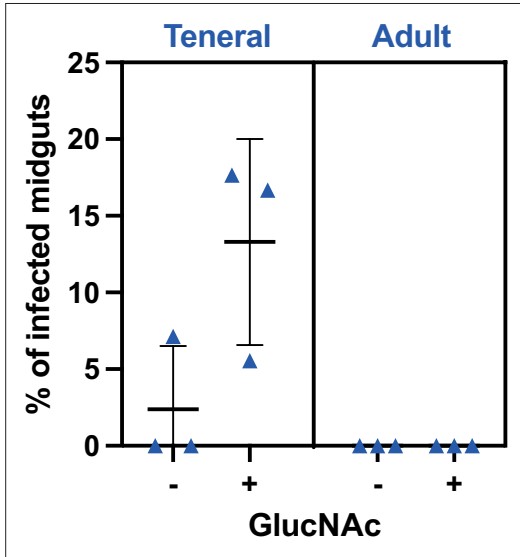

**Figure 2.** *N*-acetyl-glucosamine promotes trypanosome infection in teneral flies. Comparison of midgut infection rates in teneral vs. adult tsetse flies (blue triangles indicate batches of 50 flies, N=3 biological replicates) infected with 1000–10,000 slender parasites with (+) or without (-) *N*-acetyl-glucosamine (GlucNAc) supplement in the infective meal containing $10^5$ slender parasites/ml (equivalent to 1000–10,000 slender parasites per meal).

experimental challenge was performed to compare infection rates in teneral vs. adult flies, with or without *N*-acetyl-glucosamine supplement in an infective meal containing $10^5$ slender parasites/ml (*Figure 2*). Whereas no infection was detected in adult flies, the *N*-acetyl-glucosamine supplementation of the infective meal led to an increase in the infection rates from 2.4% to 13.3% in teneral flies (*Figure 2*).

## Discussion

The findings of *Schuster et al., 2021*, have opened a debate on the traditional view of the trypanosome life cycle where slender trypanosomes are considered as noncompetent for cyclical development in the insect vector (*Guegan and Figueiredo, 2021*). The authors proposed that their observations could provide a solution to a long-lasting paradox, namely the successful transmission of parasites in chronic infections, despite low parasitemia. Nonetheless, Schuster et al. performed all their experimental infections in laboratory conditions that were optimized for maximum transmission efficiency, with the use of teneral flies, and for some experiments, the addition of *N*-acetylglucosamine or glutathione (*Schuster et al., 2021*).

Teneral flies are young flies that remain unfed for up to 3–5 days after emergence from their puparium. They are known to be significantly more susceptible to trypanosome infection as multiple studies demonstrated they are immunologically immature (weak immune system and leaky peritrophic matrix) (*Wijers, 1958*; *Walshe et al., 2011*; *Aksoy et al., 2003*; *Weiss et al., 2013*). According to capture-recapture studies (*Hargrove, 1990*), teneral flies, however, represent a minority of individuals in wild tsetse populations. Hence, knowing that adults can live up to 9 months (*Challier, 1982*), the impact of teneral flies on trypanosome transmission may be limited, if not incidental.

In addition, Schuster et al. supplemented most infective meals with 60 mM *N*-acetylglucosamine, an inhibitor of tsetse midgut lectins (*Peacock et al., 2006*) that was also confirmed to enhance trypanosome infectivity in teneral flies in the present study. For infections with monomorphic parasites, the addition of 12.5 mM glutathione, an antioxidant that reduces the midgut environment, protected trypanosomes from cell death induced by reactive oxygen species (*MacLeod et al., 2007*). In total, these two chemical compounds used in *Schuster et al., 2021*, have inhibited the immune response in teneral flies and substantially enhanced the chances for slender trypanosomes to develop in the insect vector.

Conditions are far less favorable in the field; hence, we investigated and compared the infection potential of slender and stumpy forms in adult and teneral flies without the addition of chemicals. We observed that slender forms were not infective to adult tsetse flies and only at densities higher than $10^5$ parasites/ml. Nevertheless, in endemic areas, especially in Western Africa, parasitemia in confirmed cases is usually very low (<$10^4$ parasites/ml in Guinea, for instance), and it is necessary to concentrate parasites in blood prior to microscopic examination to increase the sensitivity of parasitological diagnosis (*Büscher et al., 2017*). Hence, the possible transmission of a few slender trypanosomes from the blood of individuals with a chronic infection is unlikely to explain the maintenance of the parasite circulation in tsetse populations.

By contrast, we observed that as few as 10 stumpy parasites are enough to produce mature infections in both teneral and adult flies, already with a significant efficiency. In patients with low parasitemia, the quorum-sensing-triggered differentiation of slender to stumpy forms could be compatible with, or even more adapted to, extravascular forms present in some tissues and organs, with a limited dilution of the parasites and parasite factors remaining concentrated locally. Indeed, extravascular PAD1-positive trypanosomes were detected in high numbers at least in adipose tissues (*Trindade et al., 2016*) and in the dermis (*Capewell et al., 2016*) of experimentally infected mice. The presence of PAD1-positive extravascular trypanosomes was also assessed in the skin of confirmed gambiense HAT cases and unconfirmed seropositive individuals in endemic areas (*Camara et al., 2021*) (and unpublished data). This suggests that stumpy trypanosomes accessible to tsetse flies are likely more abundant than previously estimated in individuals with low parasitemia.

Schuster et al. have demonstrated the intrinsic capacity of slender form trypanosomes to infect young and naive tsetse flies, highlighting the remarkable plasticity and adaptability of these protists. The fine understanding of the underlying cellular mechanisms and/or transient adaptations involved in this process remains an exciting challenge. This event is, however, unlikely to contribute to the epidemiology of African trypanosomiases in natural settings. According to both experimental and

field observations, stumpy forms appear to be the most adapted forms for African trypanosome transmission from the mammalian host to the tsetse fly vector in natural conditions.

## Materials and methods

### Strains, culture, and in vitro differentiation

The AnTat 1.1E Paris pleomorphic strain of *T. b. brucei* was derived from a strain originally isolated from a bushbuck in Uganda in 1966 (*Le Ray et al., 1977*). Bloodstream form trypanosomes were cultivated in HMI-9 medium supplemented with 10% (vol/vol) FBS (*Hirumi and Hirumi, 1989*) at 37°C in 5% $CO_2$. Proliferative slender cells were maintained at densities lower than $5 \cdot 10^5$ parasites/ml to prevent their natural quorum-sensing-dependent differentiation into stumpy forms. For in vitro slender to stumpy BSF differentiation, we used 8-pCPT-2′-*O*-Me-5′-AMP, a nucleotide analogue of 5′-AMP (BIOLOG Life Science Institute, Germany). Briefly, $2 \times 10^6$ pleomorphic AnTat 1.1E slender forms were incubated with 8-pCPT-2′-*O*-Me-5′-AMP (5 µM) for 48 hr (*Barquilla et al., 2012*). Freshly differentiated stumpy forms and slender cells were then centrifuged at 1400×*g* for 10 min and resuspended at the appropriate densities in SDM-79 medium supplemented with 10% FBS. Cells were resuspended at either $10^3$, $10^4$, or $10^5$ parasites/ml. Assuming individual blood meal volumes ranging between 10 and 100 µl, this would correspond to ingestions of 10–100, 100–1000, or 1000–10,000 parasites per condition.

### Tsetse fly maintenance, infection, and dissection

*Glossina morsitans morsitans* tsetse flies were maintained in Roubaud cages at 27°C and 70% hygrometry and fed through a silicone membrane with fresh mechanically defibrinated sheep blood (BCL, France). Adult (between 2 and 3 weeks after emergence) or teneral males (between 24 and 72 hr post-emergence) were allowed to ingest parasites through a silicone membrane. No chemical supplement was used in the first set of experiments. For assessing the effect of immunomodulatory compounds in the second set of experiments, 60 mM *N*-acetylglucosamine was added to the infective meal. A total of three to five independent biological replicates per condition were performed with batches of 50 flies per condition.

Flies were starved for at least 24 hr before being dissected blindly 28–31 days post-ingestion for isolation of all stages from the midgut and salivary glands. For recovery of all tsetse organs, after rapid isolation of the salivary glands in a first drop of phosphate-buffered saline (PBS), whole tsetse alimentary tracts, from the distal part of the foregut to the Malpighian tubules, were dissected and arranged lengthways in another drop of PBS as previously described (*Rotureau et al., 2011*; *Rotureau et al., 2012*). Isolated organs were then scrutinized under a microscope at ×40 magnification by two independent readers, and infection rates per organ were scored (*Rotureau et al., 2014*).

### Immunofluorescence analysis

Cultured parasites were washed in TDB and spread onto poly-L-lysine-coated slides. For flash methanol fixation, slides were air-dried for 10 min, fixed in methanol at –20°C for 5 s, and rehydrated for 20 min in PBS. For immunodetection of stumpy forms, slides were incubated for 1 hr at 37°C with a rabbit polyclonal anti-PAD1 antibody (kindly provided by Keith Matthews, University of Edinburgh) (*Dean et al., 2009*) diluted at 1:300 in PBS containing 0.1% bovine serum albumin (BSA). After three consecutive 5 min washes in PBS, a species and subclass-specific secondary antibody coupled to the Alexa 488 fluorochrome (Jackson ImmunoResearch) diluted at 1:1000 in PBS containing 0.1% BSA was applied for 1 hr at 37°C. After washing in PBS, slides were finally stained with 4',6-diamidino-2-phenylindole (1 µg/ml) for visualization of kinetoplast and nuclear DNA content and mounted under coverslips with ProLong Antifade Reagent (Invitrogen), as previously described (*Rotureau et al., 2011*). Slides were observed under an epifluorescence DMI4000 microscope (Leica) with a ×100 oil objective (NA 1.4) to assess the proportion of PAD1-positive cells in the infective meals (n>100 cells/condition).

### Statistical analysis

Infections rates were compared by a two-sided ANOVA at 95% confidence with Prism V10.0.3 (GraphPad). MG infection rate comparisons were statistically significant between teneral and adult flies infected with ST in each amount (p<0.02 with 10 parasites; p<0.0001 with 100 and 1000 parasites) and with 1000 SL (p<0.0001). MG infection rate comparisons were statistically significant (p<0.0001)

between parasite stages (SL and ST) in each amount (10, 100, and 1000) and for each fly group (teneral and adult), except in teneral flies infected with 1000 parasites (p=0.2356).

## Acknowledgements

We thank K Matthews for providing the anti-PAD1 antibody and P Bastin for his critical reading of the manuscript. This work was supported by the Institut Pasteur, the Programme Investissement d'Avenir of the French Government, Laboratoire d'Excellence, ANR-10-LABX-62-IBEID and ANR-11-LABX-0024-PARAFRAP. This work was supported by the French National Agency for Scientific Research projects ANR-18-CE15-0012 TrypaDerm and ANR-19-CE15-0004-02 AdipoTryp. None of these funding sources has a direct scientific or editorial role in the present study.

## Additional information

### Funding

| Funder | Grant reference number | Author |
|---|---|---|
| Agence Nationale de la Recherche | ANR-10-LABX-62-IBEID | Brice Rotureau |
| Agence Nationale de la Recherche | ANR-11-LABX-0024-PARAFRAP | Parul Sharma |
| Agence Nationale de la Recherche | ANR-18-CE15-0012 TrypaDerm | Brice Rotureau |
| Agence Nationale de la Recherche | ANR-19-CE15-0004-02 AdipoTryp | Brice Rotureau |

The funders had no role in study design, data collection and interpretation, or the decision to submit the work for publication.

### Author contributions

Jean Marc Tsagmo Ngoune, Parul Sharma, Data curation, Formal analysis, Investigation, Writing – review and editing; Aline Crouzols, Nathalie Petiot, Investigation; Brice Rotureau, Conceptualization, Resources, Data curation, Formal analysis, Supervision, Funding acquisition, Validation, Visualization, Methodology, Writing – original draft, Project administration, Writing – review and editing

### Author ORCIDs

Brice Rotureau ⬥ https://orcid.org/0000-0003-0671-8999

Reviewer #2 (Public review): https://doi.org/10.7554/eLife.91602.4.sa1
Reviewer #3 (Public review): https://doi.org/10.7554/eLife.91602.4.sa2
Author response https://doi.org/10.7554/eLife.91602.4.sa3

## Additional files

### Supplementary files

MDAR checklist

### Data availability

All data generated and analysed during this study are included in the manuscript and supporting files.

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
