## [Editor Report · eLife Assessment]

African (or Salivarian) trypanosomes are significant pathogens of humans and domestic animals. For many decades is was accepted that only the "stumpy" non-proliferative form was capable of infecting the Tsetse-fly vector, but recent work challenged this, suggesting that the proliferative "slender" form is also infective. The current paper provides **important** and **convincing** laboratory evidence that the original concept is probably correct for most infections: the slender form was not infective for adult Tsetse, and was only able to infect young, less immunocompetent flies if N-acetyl glucosamine was added to the feed.

---

## [Referee Report · Reviewer #2 (Public review)]

Summary:

In contrast to the recent findings reported by Schuster S et al., this brief paper presents evidence suggesting that the stumpy form of *T. brucei* is likely the most pre-adapted form to progress through the life cycle of this parasite in the tsetse vector.

Strengths:

One significant experimental point is that all fly infection experiments are conducted in the absence of "boosting" metabolites like GlcNAc or S-glutathione. As a result, flies infected with slender trypanosomes present very low or nonexistent infection rates. This provides important experimental evidence that the findings of Schuster S and colleagues may need to be revisited.

In the revised submission the authors also compared trypanosome midgut infection levels in tsetse flies when either young (teneral) or mature adult flies received infected bloodmeals, with or without 60 mM GlcNAc. The data clearly show that, unlike in teneral flies, the addition of GlcNAc to the trypanosome-infected bloodmeal does not enhance midgut infection in mature adult flies. This is now convincingly demonstrated in Figure 2 and provides strong experimental support for the suggestion that the effect reported by Schuster S. et al. may have been influenced by both fly age and the inclusion of metabolic "boosters" in the bloodmeal.

---

## [Referee Report · Reviewer #3 (Public review)]

The dogma in the Trypanosome field is that transmission by Tsetse flies is ensured by stumpy forms. This has been recently challenged by the Engstler lab (Schuster et al.), who showed that slender forms can also be transmitted by teneral flies. In this work, the authors aimed to test whether transmission by slender forms is possible and frequent. The authors observed that most stumpy forms infections with teneral and adult flies were successful while only 1 out of 24 slender form infections were successful.

The comparison of midgut infection in adult vs teneral flies was significant in most of the conditions. However, the critical comparison is still missing: within each type of fly (adult or teneral), was the MG infection significantly different between slender and stumpy forms?

Figure 2 convincingly demonstrates the effect of the metabolite N-acetylglucosamine on Tsetse infection. This addition helps better integrate the study with previous work. I thank the authors for their effort in performing this experiment.

It is still remains unknown why this work and Schuster et al. reached different conclusions. As a result it remains unclear in which conditions slender forms could be important for transmission. Several variables could explain differences between the two groups: the strain used, the presence or absence of glutathione, how Tsetse colonies were maintained, thorough molecular and cellular characterisation of slender and stumpy forms (to avoid using intermediate forms as slender forms), comparison to recent field parasite strains.

---

## [Author Response]

The following is the authors’ response to the previous reviews

**eLife Assessment**
For decades it has been accepted that only the growth-arrested "stumpy" form of *Trypanosoma brucei* can infect the arthropod vector, the Tsetse fly, but this was recently challenged by a demonstration that - under artificial conditions that are known to enhance infectivity - the proliferative "slender" form can also establish Tsetse infections. The infectiousness of the two forms is a fundamental question in trypanosome biology and epidemiology, concerning both infection dynamics and parasite differentiation. The authors of the current study provide compelling evidence that without artificial enhancement, the "stumpy" form is indeed much more infective for Tsetse than the slender form; they suggest that this is probably also true in the wild.Since the authors of this paper did not themselves test the effect of enhancing conditions, the precise reason for the discrepancy in results between the two laboratories has not been demonstrated conclusively.

This specific comment was addressed in the revision and illustrated with new data.

Differences between the strain clones, the cell culture conditions and/or the fly colony maintenance conditions could explain part of the differences in infection rates observed here as compared to the Schuster et al. study (1). However, the use of the lectin-inhibitory sugar N-acetyl-glucosamine to enhance infection rates in the latter study could be a more likely explanation. To assess this hypothesis, an additional experimental challenge was performed to compare infection rates in teneral versus adult flies, with or without N-acetyl-glucosamine supplement in an infective meal containing 10^5^ slender parasites / ml (Figure 2). Whereas no infection was detected in adult flies, the N-acetyl-glucosamine supplementation of the infective meal led to an increase of the infection rates from 2,4% to 13,3% in teneral flies (Figure 2).

**Public Reviews:**

**Reviewer #1 (Public Review):**
Summary:Ngoune et al. present compelling evidence that Slender cells are challenged to infect tsetse flies. They explore the experimental context of a recent important paper in the field, Schuster et al., that presents evidence suggesting the proliferative Slender bloodstream *T.brucei* can infect juvenile tsetse flies. Schuster et al. was disruptive to the widely accepted paradigm that the Stumpy bloodstream form is solely responsible for tsetse infection and *T. brucei* transmission potential. Evidence presented here shows that in all cases, Stumpy form parasites are exponentially more capable of infecting tsetse flies. They further show that Slender cells do not infect mature flies.However, they raise questions of immature tsetse immunological potential and field transmission potential that their experiments do not address. Specifically, they do not show that teneral tsetse flies are immunocompromised, that tsetse flies must be immunocompromised for Slender infection nor that younger teneral tsetse infection is not pertinent to field transmission.

All these specific comments were addressed in the revision and illustrated with new data and references.

- The limited immunocompetence of teneral flies has been extensively studied by the labs of S. Aksoy at Yale and M. Lehane at Liverpool. In the discussion, we provide key references from these two labs 19-22.

- Differences between the strain clones, the cell culture conditions and/or the fly colony maintenance conditions could explain part of the differences in infection rates observed here as compared to the Schuster et al. study (1). However, the use of the lectin-inhibitory sugar N-acetyl-glucosamine to enhance infection rates in the latter study could be a more likely explanation. To assess this hypothesis, an additional experimental challenge was performed to compare infection rates in teneral versus adult flies, with or without N-acetyl-glucosamine supplement in an infective meal containing 10^5^ slender parasites / ml (Figure 2). Whereas no infection was detected in adult flies, the N-acetyl-glucosamine supplementation of the infective meal led to an increase of the infection rates from 2,4% to 13,3% in teneral flies (Figure 2).

- Our comment on the relevance to field transmission is simply based on field observations of the fly biology. For example, according to the capture-recapture experiments described in HARGROVE JW insect sci applic 1990 (new ref 23), wild female mortality was reported 6.8% shortly after emergence, <1% for ages 20-50 days and rose to 5% by 130 day (a pattern similar to that for laboratory reared tsetse), while wild male daily mortality was 8.3% after emergence, fell to 5.5% by 9 days, then rose continuously to more than 10% by 30 days. This means that adult flies represent the majority of individuals in a wild tsetse population. Hence, knowing that both males and females are strictly hematophagous and that they can live up to nine months, the impact of teneral flies (up to 4 days after emergence) on trypanosome transmission appears limited, if not incidental.

Strengths:Experimental Design is precise and elegant, outcomes are convincing. Discussion is compelling and important to the field. This is a timely piece that adds important data to a critical discussion of host:parasite interactions, of relevance to all parasite transmission.

Thank you

Weaknesses:As above, the authors dispute the biological relevance of teneral tsetse infection in the wild, without offering evidence to the contrary. Statements need to be softened for claims regarding immunological competence or relevance to field transmission.

All these specific comments were addressed in the revision and illustrated with new data and references.

- The limited immunocompetence of teneral flies has been extensively studied by the labs of S. Aksoy at Yale and M. Lehane at Liverpool. In the discussion, we provide key references from these two labs 19-22.

- Differences between the strain clones, the cell culture conditions and/or the fly colony maintenance conditions could explain part of the differences in infection rates observed here as compared to the Schuster et al. study (1). However, the use of the lectin-inhibitory sugar N-acetyl-glucosamine to enhance infection rates in the latter study could be a more likely explanation. To assess this hypothesis, an additional experimental challenge was performed to compare infection rates in teneral versus adult flies, with or without N-acetyl-glucosamine supplement in an infective meal containing 10^5^ slender parasites / ml (Figure 2). Whereas no infection was detected in adult flies, the N-acetyl-glucosamine supplementation of the infective meal led to an increase of the infection rates from 2,4% to 13,3% in teneral flies (Figure 2).

- Our comment on the relevance to field transmission is simply based on field observations of the fly biology. For example, according to the capture-recapture experiments described in HARGROVE JW insect sci applic 1990 (new ref 23), wild female mortality was reported 6.8% shortly after emergence, <1% for ages 20-50 days and rose to 5% by 130 day (a pattern similar to that for laboratory reared tsetse), while wild male daily mortality was 8.3% after emergence, fell to 5.5% by 9 days, then rose continuously to more than 10% by 30 days. This means that adult flies represent the majority of individuals in a wild tsetse population. Hence, knowing that both males and females are strictly hematophagous and that they can live up to nine months, the impact of teneral flies (up to 4 days after emergence) on trypanosome transmission appears limited, if not incidental.

**Reviewer #2 (Public Review):**
Summary:In contrast to the recent findings reported by Schuster S et al., this brief paper presents evidence suggesting that the stumpy form of *T. brucei* is likely the most pre-adapted form to progress through the life cycle of this parasite in the tsetse vector.Strengths:One significant experimental point is that all fly infection experiments are conducted in the absence of "boosting" metabolites like GlcNAc or S-glutathione. As a result, flies infected with slender trypanosomes present very low or nonexistent infection rates. This provides important experimental evidence that the findings of Schuster S and colleagues may need to be revisited.

Thank you

Weaknesses:However, I believe the authors should have included their own set of experiments demonstrating that the presence of these metabolites in the infectious bloodmeal enhances infection rates in flies receiving blood meals containing slender trypanosomes. Considering the well-known physiological variabilities among flies from different facilities, including infection rates, this would have strengthened the experimental evidence presented by the authors.

This specific comment was addressed in the revision and illustrated with new data.

Differences between the strain clones, the cell culture conditions and/or the fly colony maintenance conditions could explain part of the differences in infection rates observed here as compared to the Schuster et al. study (1). However, the use of the lectin-inhibitory sugar N-acetyl-glucosamine to enhance infection rates in the latter study could be a more likely explanation. To assess this hypothesis, an additional experimental challenge was performed to compare infection rates in teneral versus adult flies, with or without N-acetyl-glucosamine supplement in an infective meal containing 10^5^ slender parasites / ml (Figure 2). Whereas no infection was detected in adult flies, the N-acetyl-glucosamine supplementation of the infective meal led to an increase of the infection rates from 2,4% to 13,3% in teneral flies (Figure 2).

**Reviewer #3 (Public Review):**
The dogma in the Trypanosome field is that transmission by Tsetse flies is ensured by stumpy forms. This has been recently challenged by the Engstler lab (Schuster et al.), who showed that slender forms can also be transmitted by teneral flies. In this work, the authors aimed to test whether transmission by slender forms is possible and frequent. The authors observed that most stumpy forms infections with teneral and adult flies were successful while only 1 out of 24 slender form infections were successful.In this revised version of the manuscript, the authors made some text changes and included statistical testing as a new section of the Materials and Methods. It seems the comparison of midgut infection in adult vs teneral flies was significant in most of the conditions. However, the critical comparison is still missing: within each type of fly (adult or teneral), was the MG infection significantly different between slender and stumpy forms?

An ANOVA statistical analysis was performed and a dedicated section added to the revised version. MG infection rate comparisons were statistically significant between teneral and adult flies infected with ST in each amount (p<0.02 with 10 parasites; p<0.0001 with 100 and 1,000 parasites) and with 1,000 SL (p<0.0001). MG infection rate comparisons were statistically significant (p<0.0001) between parasite stages (SL and ST) in each amount (10, 100 and 1,000) and for each fly group (teneral and adult), excepted in teneral flies infected with 1,000 parasites (p=0.2356).

Given no additional experiments were performed, it remains unknown why this work and Schuster et al. reached different conclusions. As a result it remains unclear in which conditions slender forms could be important for transmission. Several variables could explain differences between the two groups: the strain used, the presence or absence of N-acetylglucosamine and/or glutathione, how Tsetse colonies were maintained, thorough molecular and cellular characterisation of slender and stumpy forms (to avoid using intermediate forms as slender forms), comparison to recent field parasite strains.

This specific comment was addressed in the revision and illustrated with new data.

Differences between the strain clones, the cell culture conditions and/or the fly colony maintenance conditions could explain part of the differences in infection rates observed here as compared to the Schuster et al. study (1). However, the use of the lectin-inhibitory sugar N-acetyl-glucosamine to enhance infection rates in the latter study could be a more likely explanation. To assess this hypothesis, an additional experimental challenge was performed to compare infection rates in teneral versus adult flies, with or without N-acetyl-glucosamine supplement in an infective meal containing 10^5^ slender parasites / ml (Figure 2). Whereas no infection was detected in adult flies, the N-acetyl-glucosamine supplementation of the infective meal led to an increase of the infection rates from 2,4% to 13,3% in teneral flies (Figure 2).

**Recommendations for the authors:**

**Reviewer #1 (Recommendations For The Authors):**
The manuscript is improved, but the author has not addressed much of the constructive criticism offered that would benefit the manuscript.To clarify, evidence from Schuster et al did not demonstrate, rather it suggested. That is a major point of this paper - that the previous evidence presented had caveats. Terms such as demonstrate or prove are inappropriate in most biological contexts, unless evidence is without caveat.

This specific comment was addressed in the revision and illustrated with new data.

Differences between the strain clones, the cell culture conditions and/or the fly colony maintenance conditions could explain part of the differences in infection rates observed here as compared to the Schuster et al. study (1). However, the use of the lectin-inhibitory sugar N-acetyl-glucosamine to enhance infection rates in the latter study could be a more likely explanation. To assess this hypothesis, an additional experimental challenge was performed to compare infection rates in teneral versus adult flies, with or without N-acetyl-glucosamine supplement in an infective meal containing 10^5^ slender parasites / ml (Figure 2). Whereas no infection was detected in adult flies, the N-acetyl-glucosamine supplementation of the infective meal led to an increase of the infection rates from 2,4% to 13,3% in teneral flies (Figure 2).

Statements regarding teneral flies in the field are softened. Yet the referenced papers pertain more to commensurate coinfections rather than reduced immunocapacity of immature teneral flies in the field. This should be clarified.

The limited immunocompetence of teneral flies has been extensively studied by the labs of S. Aksoy at Yale and M. Lehane at Liverpool. In the discussion, we provide key references from these two labs 19-22.

The text remains convoluted to read with grammatical errors in places. For example, it is incorrect to begin a sentence with However. There are far too many run-on sentences in the manuscript that confuse this straightforward story.

The revised text was improved as much as possible.

All text requires grammatical refinement and softer claims unless additional experiments are undertaken.
**Reviewer #2 (Recommendations For The Authors):**
I continue to endorse the publication of this manuscript; however, I am somewhat disappointed by the authors' justifications for not conducting additional experiments or exploring other factors that might influence the infection phenotypes in the fly.

This specific comment was addressed in the revision and illustrated with new data.

Differences between the strain clones, the cell culture conditions and/or the fly colony maintenance conditions could explain part of the differences in infection rates observed here as compared to the Schuster et al. study (1). However, the use of the lectin-inhibitory sugar N-acetyl-glucosamine to enhance infection rates in the latter study could be a more likely explanation. To assess this hypothesis, an additional experimental challenge was performed to compare infection rates in teneral versus adult flies, with or without N-acetyl-glucosamine supplement in an infective meal containing 10^5^ slender parasites / ml (Figure 2). Whereas no infection was detected in adult flies, the N-acetyl-glucosamine supplementation of the infective meal led to an increase of the infection rates from 2,4% to 13,3% in teneral flies (Figure 2).